# Dynamics of international Trade: A 30-year analysis of key exporting nations

**Nobuo Yazawa** [ID] *

Faculty of International Business Management, Beppu University, Beppu, Oita, Japan

* yazawa@nm.beppu-u.ac.jp

**Data Availability Statement:** All evidence data are available from the Zenodo database (URL: https://zenodo.org/record/7792299#.ZCkUbS_3KjQ DOI: 10.5281/zenodo.7792299).

## Abstract

This article aims to analyze the evolution of trading networks, emphasizing aspects of centrality and reciprocity among the major exporting nations, specifically, the U.S., China, India, Japan, and South Korea, from 1992 to 2020. The study problem we address is how these network structures have shifted over time, and what the implications of these changes are for international economic relations and policy. We further consider the impact of major global events on these trade networks and how they have shaped the evolution of these networks. We utilized three distinct methods. First, we examined time-series trade data during the study period and quantified network reciprocity through the sum of squared trade imbalances across different product categories. Second, we visualized these trade networks using arrows, with their sizes being proportional to the trade value between each pair of countries; significant trade relationships were indicated by arrows with a standard deviation value of 55 or above. Finally, we introduced a new cluster analysis methodology for studying the evolution of network structures over time. This method utilized an 80-dimensional vector representation of the annual networks, divided into four categories, and the resulting structures were visualized as dendrograms using R software. The network structure has become more reciprocal for most product categories, and the center of the network has shifted from the U.S. to China for all product categories, except for consumer goods and raw materials. The study also highlights the significant impact of global events and crises such as 9/11 attacks on the international trade network structure. Our findings inform several policy recommendations. These include encouraging balanced trade for economic stability and improved international relations, realigning trade focus in response to the shift in trade network center, and developing resilience policies that account for the substantial impact of global events on trade networks.

## 1. Introduction

The motivation for this study lies in the quest to comprehend international trade from a network perspective. Although international trade has been extensively studied, the network approach remains relatively uncharted territory. We posit that this perspective could provide novel insights into the dynamics and resilience of trade relationships amidst global challenges,

**Funding:** The author received no specific funding for this work.

**Competing interests:** The authors have declared that no competing interests exist.

emphasizing the interconnectedness of worldwide trade. This study will augment the existing literature by offering a fresh lens to view international trade dynamics.

Our objective is to examine the evolution in structural properties, including centrality and reciprocity, of trade networks among the United States (U), China (C), India (I), Japan (J), and South Korea (K) from 1992 to 2020. The significance of our work lies in its unique perspective on international trade dynamics, focusing on an underexplored network approach. By investigating the interconnectedness of global trade, we aim to offer novel insights into the resilience of trade relationships amidst global challenges and the fluidity of these dynamics.

The theoretical framework of our paper integrates the following components:

1. International Trade Theory: This theory serves as the bedrock for comprehending and analyzing international trade dynamics. It is employed in our exploration of evolving trade relationships among U, C, I, J, and K over 29 years. The theory aids in interpreting trade imbalances, shifts in trade dominance, and their responses to various political and economic crises.

2. Network Analysis: This methodological approach enables us to scrutinize the structure of international trade relationships. Network analysis aids in visualizing and understanding intricate relationships between trade partners, providing insights into the concepts of centrality and reciprocity. We adopt hierarchical clustering as a technique for community detection in network theory.

3. Time-Series Analysis: This analytical technique is used to examine data points collected at regular time intervals. In our study, we apply time-series analysis to reveal how trade networks' structure has evolved over time and to assess the impact of significant events, such as political and economic crises.

4. Crisis Theory: We use this theory to gauge the impact of political and economic crises on international trade networks. It provides an analysis of how these crises disrupt trade flows, alter trade networks' structure, and shift the centrality and reciprocity of trading partners.

Our methodological approach involves gathering trade data between the five countries over the 29-year period, utilizing network analysis tools to interpret the structure and dynamics of their trade relationships. This is supplemented by a time-series analysis to pinpoint trends and patterns over time, and an application of crisis theory to understand the influence of political and economic crises on these trade networks.

By adopting a systematic approach, our study examines the structural characteristics of international trade networks. Leveraging international trade theory, we delve into the dynamics of trade relationships over a 29-year period. Through network analysis, we visualize these relationships, understanding their centrality and reciprocity. Time-series analysis helps us identify trends and changes over time, while crisis theory enlightens us on the effects of significant political and economic crises on these networks. We blend these theoretical perspectives and methodological tools to present a holistic analysis of international trade dynamics.

## 2. Literature review

International relations have grown increasingly complex since the 1970s, and economic interdependence between nations is deepening [1]. This interconnectedness has resulted from technological advances that have reduced transportation and communication costs, leading to economic globalization [2].

However, some researchers have observed a trend of de-globalization, with economic interconnectedness decreasing at the global level in terms of international business, trade, and foreign direct investment [3].

Several factors have contributed to changes in the interdependency of world trade, including global economic crises, anti-dumping policies [4], and voluntary export reductions [5]. The C-U trade imbalance is a particularly pressing international issue, partly due to the low-

value-added content of C's exports [6]. Madhur [7] emphasized the importance of trilateral economic cooperation among C, J, and K, while others have warned of the potential dangers of a more powerful C [8]. Nonetheless, the WTO, in cooperation with the IMF, has established a transparent system of conflict resolution for trade frictions and contributed to the development of global trade [9].

Our study sets itself apart from existing literature through its unique methodological approach and the ensuing findings. Garlaschelli and Loffredo [10] centered their research on link reciprocity in binary directed networks, employing the correlation coefficient between adjacency matrix entries of a directed graph to formulate a new reciprocity definition. We too analyzed reciprocity, but assessed it in a weighted directed network, gauged by the sum of squared trade imbalances between actor pairs. This method simplifies reciprocity measurement compared to matrix computation methods.

In the study "A Network of Networks Perspective on Global Trade" (2015) [11], researchers dissected 186 national economies into 26 industry sectors using multi-regional input-output data. They detected anomalies in trade patterns using measures like the Hamming distance between the International Trade Network (ITN) in the current and preceding year. Contrarily, our research applies clustering analysis to time vectors. We adopted this method because it was innovative to use clustering analysis, commonly used for grouping points distributed in space, to group points distributed over time.

The 2022 paper "The rise and fall of countries in the global value chains" [12] employed graph theory's eigenvector centrality to gauge node importance in a network, noting a significant role reversal between the US and China in 2007. Our research, however, visualizes time series progression of network structure, observing the network center shifting from U to C in 2007 for all products, in 2005 for capital goods, and in 2006 for intermediate goods. This method's selection is due to visualization's intrinsic ability to communicate simply and clearly, compared to complex methods involving eigenvalue matrix computations.

Our research makes distinct contributions to the literature. Firstly, our focus on network reciprocity, measured by the sum of squared trade imbalances, offers a fresh lens to scrutinize the global trade network. This new method helps discern reciprocity trends among various product categories. Secondly, visualizing the time series progression of network structure provides insights into the shifting network center, complementing the eigenvector centrality measure used in previous studies. Lastly, applying clustering analysis to time vectors introduces a novel method to study alterations in the global trade network. This innovative approach indicates that half of the significant changes in network structure occur during considerable crises, highlighting global trade's vulnerability to such events. Overall, our study enriches the current comprehension of global trade networks by introducing new methodologies and offering fresh insights into their structure and dynamics (Table 1).

Our study contributes to the literature in a few distinct ways. Firstly, our focus on the degree of reciprocity within a network measured by the sum of squared trade imbalances provides a new perspective to examine the global trade network. This novel method helps identify a divergence in reciprocity trends among different product categories. Secondly, the visualization of the time series progression of network structure offers insights into the shifting center of the network. This is a unique angle that complements the eigenvector centrality measure used in prior studies. Thirdly, our use of clustering analysis applying to time vectors introduces a new method of studying changes in the global trade network. This innovative approach suggests that half of the substantial alterations in the network structure occur during significant crises, pointing to the susceptibility of global trade to these events. Overall, our work expands on the current understanding of global trade networks, introducing new methodologies and providing fresh insights into their structure and dynamics.

**Table 1. Literature table.**

| Articles | Their method | Their key findings | Our methods | Our key findings |
|---|---|---|---|---|
| [12] | An eigenvector centrality of graph theory was applied as a common measure for assessing the importance of nodes in a network | 2007 marked an inflection point at which new winners and losers emerged and a remarkable reversal of leading role took place between the two major economies, the US and China. | Visualization of the time series progression of network structure with arrows whose standard deviation are 55 or greater. | The center of the network has shifted from U to C in 2007 for all products, 2005 for capital goods, 2006 for intermediate goods. No shift for consumer goods and raw materials. |
| [10] | The study of link reciprocity in binary directed networks of global trade<br>a definition of reciprocity as the correlation coefficient between the entries of the adjacency matrix of a directed graph year: 1948–2000 | Reciprocity of trade network increased from 0.68 (1948) to 0.9 (2000) | The degree of reciprocity within a network measured by the sum of squared trade imbalances between each pair of actors | the (overall increase in squared trade imbalances) decrease of reciprocity for all products and consumer goods, but the (decrease in the squared trade imbalances) increase of reciprocity for capital goods, intermediate goods, and raw materials. |
| [11] | multi-regional input-output data to decompose 186 national economies into 26 industry sectors 1990–2010<br>(a) a definition of reciprocity as $r = \frac{1}{|A|} Tr[A]^2$ where A is an adjacency matrix | (a) The reciprocity (r) gradually increases in the national partition Cc, but saturating in 2000. | | |
| | (b) measure the Hamming distance between the international trade network in the present and the preceding year | (b) Hm is an applicable measure to identify anomalies in trade patterns, such as the financial crisis in 2009 | Clustering analysis applying to time vectors | That the half of the substantial alterations in the network structure occur during such incidents suggests that global trade is significantly susceptible to these crises. |

## 3. Materials and methods

Suppose there are n actors A1, A2, . . ., An. When a link from A1 to A2 and a link from A2 to A1 exist, and both links have the same flow quantity, we say the relation between A1 and A2 is perfectly mutual (Fig 1(A)). On the other hand, when the link from A1 to A2 has a large flow

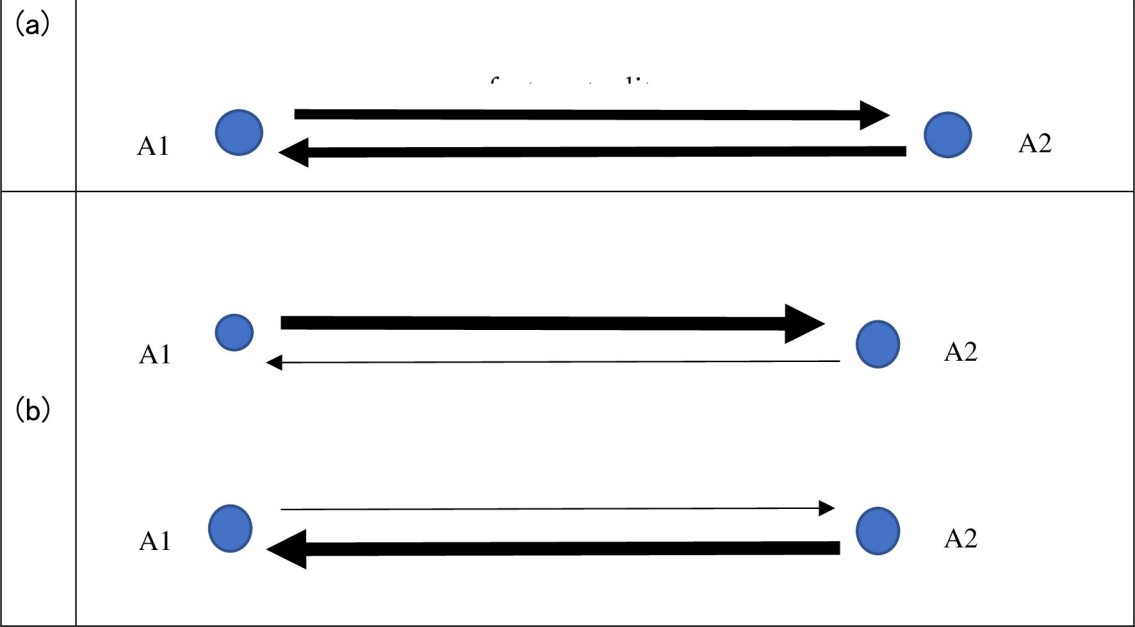

**Fig 1. Three types of mutuality between two actors.**

quantity and the link from A2 to A1 has a little flow quantity, we say the relation between A1 and A2 has little mutuality (Fig 1(B)).

We will quantify the strength of the link and let it have positive real number a. Let the flow quantity of the link from A1 to A2 be F(1,2), and from A2 to A1 be F(2,1). Then we quantify the proportion of the flow from Ai to Aj in the total flow of the network as $X(i, j, t_l)$ in the following way.

In this subsection, we regard directed links as output flows from actors. We define $X(1,2, t_l)$ representing the proportion of output flow from A1 to A2 in the total flow in the network at the time $t_l$ in Eq (1).

$$X(1, 2, t_l) = \frac{F(1, 2, t_l)}{T(t_l)} \tag{1}$$

$T(t_l)$ is defined in Eq (2)

$$T(t_l) \equiv \sum_{i=1}^{n} \sum_{k \neq i} F(i, k, t_l) \tag{2}$$

In general, $X(i, j, t_l)$ is formulated as below (Eq (3)).

$$X(i, j, t_l) \equiv \frac{F(i, j, t_l)}{T(t_l)} \tag{3}$$

In Eq (3), $X(i, j, t_l)$ represents the proportion of the flow from Ai to Aj in the total flow of the network at the time $t = t_l$.

$$\sum_{i=1}^{n} \sum_{j \neq i} X(i, j, t_l) = 1 \tag{4}$$

## 3.1. Data collection

We chose to use the World Integrated Trade Solution (WITS) database [13] for our study for several reasons. Firstly, the WITS database is one of the most comprehensive sources of international trade data. It contains data from multiple organizations including the United Nations Comtrade database, the World Bank, and the World Trade Organization, making it a one-stop-shop for trade data. Secondly, it provides access to the most detailed level of product classification, allowing for granular and nuanced analyses. This includes data at the Harmonized System (HS) codes for traded goods, which can be very specific (e.g., different types of fruits are separately classified). Lastly, the data in WITS is sourced from reputable international organizations and is considered highly reliable.

We selected U, C, I, J, and K for our study because they are among the most significant players in international trade, contributing extensively to global export dynamics. As such, examining these countries allows for meaningful insights into global trade patterns. Regarding the selected timeline from 1992 to 2020, this range was chosen based on the data availability in the World Integrated Trade Solution (WITS) database. As of May 1, 2023, the earliest available data for the economies we are studying is from 1992, and the most recent data is from 2020. This period allows for a comprehensive, longitudinal study of the changes in trade networks over nearly three decades.

The dataset provided information on trade flows for four categories of goods, namely, capital goods, consumer goods, intermediate goods, and raw materials in thousand US dollar. The combined value of the four categories constitutes the total for all products. Firstly, capital goods are long-lasting assets used in the production of other goods and services. These goods

do not get consumed or transformed during the production process. Examples include machinery, equipment, buildings, and vehicles. Capital goods are essential for businesses to increase their productivity, and their acquisition represents an investment in the future. Secondly, consumer goods are products intended for direct consumption by end-users. These goods are typically produced for the purpose of satisfying individual wants and needs. Consumer goods can be further divided into two subcategories: (a) Durable consumer goods: These are goods with a relatively long lifespan, such as cars, appliances, and furniture. They usually have a lifespan of more than three years, (b) Non-durable consumer goods: These are goods with a short lifespan, such as food, beverages, and clothing. They are consumed quickly or have a lifespan of less than three years. Thirdly, intermediate goods are products used in the production process that get transformed or incorporated into the final product. These goods are not sold to end-users but are instead used by businesses as inputs in the production of other goods or services. Examples include steel used in car manufacturing, flour used in bread production, and electronic components used in assembling smartphones. Lastly, raw materials are natural resources or basic substances that serve as inputs in the production of goods. These materials are extracted or harvested from the environment and are often unprocessed or minimally processed before being used in production. Examples of raw materials include metals like iron ore, agricultural products like wheat and banana, and energy sources like crude oil.

## 3.2. Methods

As previously outlined in Chapter 1: Introduction, the fundamental theory guiding our analysis is as follows:

International Trade Theory: This theory undergirds our analysis of trade dynamics among the U.S. (U), China (C), India (I), Japan (J), and South Korea (K). It aids us in interpreting trade imbalances, shifts in trade dominance, and the impact of political and economic crises on these aspects. The principles of this theory guide our examination of these trade relationships over a 29-year period.

Network Analysis: This methodological approach is integral to our study as it helps us to unravel the complex structure of international trade relationships. Network Analysis, in our context, enables us to visualize and comprehend the complexities of trade networks, shedding light on their centrality and reciprocity. It allows us to identify dominant players and the characteristics of their trade relationships. Furthermore, we use hierarchical clustering as a method of network theory for community detection.

Time-Series Analysis: We employ this analytical method to examine trade data collected at regular intervals. The principles of Time-Series Analysis assist us in identifying underlying trends, cycles, and patterns in the data. This approach is crucial in our investigation of the evolution of the structure of trade networks over time and in assessing the impact of significant events such as political and economic crises.

Crisis Theory: This theory guides our analysis of how political and economic crises disrupt trade flows and reshape the structure of trade networks. It facilitates our understanding of shifts in the centrality and reciprocity of trading partners in response to crises.

In our study, we employed the following three methods:

**3.2.1 Analysis of the time series variation in trade balance between partners.** The degree of reciprocity within a trade network can be quantified by calculating the sum of squared trade imbalances between each pair of actors. Thus, from Eq (3), for each item, the following value

$$\sum_{i,j} |X(i,j,t_l) - X(j,i,t_l)|^2 \qquad (i = 1,2,3,4,5; j = 1,2,3,4,5; i \neq j) \tag{5}$$

was computed from 1992 to 2020. As this value decreases, the reciprocity within the network increases. We examined the time-series changes in the values of all products, capital goods, consumer goods, intermediate goods, and raw materials, and examined whether the bidirectional relationship has deepened.

**3.2.2. Visualization of the time series progression of network structure.** Using trade data ($X(i, j, t_l)$) from 1992 to 2020 between five countries, we constructed a network of major directed arrows between the five nodes on a PowerPoint slide. The width of the arrows is proportional to $X(i, j, t_l)$. Consequently, 29 networks were created in a time series for the items (all products, capital goods, consumer goods, intermediate goods, and raw materials). Major directed arrows refer to those arrows with a value of 55 or higher in terms of standard deviation value within the set of all directed arrows for each item for each year. Thus, links with a standard deviation of 55 or greater were depicted out of a total of 20 links on the slides.

**3.2.3. Hierarchical clustering.** Previous studies on cluster analysis have focused on countries in trade networks, whereas the present study innovatively approaches the subject by targeting 29 years, ranging from 1992 to 2020, and grouping years with similar characteristics through cluster analysis [14]. Annually, there are networks for four categories (capital goods, consumer goods, intermediate goods, and raw materials) of items, each containing 20 arrows. Therefore, the initial step involves calculating twenty nine 80-dimensional vectors ($4 \times 20 = 80$) representing the network structure. A total of 29 vectors, covering the period from 1992 to 2020, were obtained. These vectors were then subjected to hierarchical clustering using the following steps.

Step 1: Distance Matrix Creation: The distance matrix was created by calculating the Euclidean distance between the vectors of each year.

Step 2: Hierarchical Clustering: The hierarchical clustering was performed using the average linkage

method, where each vector starts as a separate cluster, and the closest pairs (years) are merged iteratively, until all vectors are in one cluster.

Step 3: Visualization: The dendrogram, a tree-like diagram, was used to visualize the clusters of

29 vector (years), allowing us to identify the sets of years each set, as a cluster, composed of neighboring years whose distance is measured by the Euclidean distance in step1.

Note that the study employed R software (version 4.0.3) for hierarchical clustering. The results of the study provide insights into the changing dynamics of international trade networks.

# 4. Analysis and results

## 4.1. Analysis of the time series variation in trade balance between partners

From Fig 2 and Table 2, the squared sum of trade imbalances for all products shows an overall increase from 0.036 in 1992 to 0.041 in 2020, despite some fluctuations in between. the highest value was recorded in 2015 and 2020 at 0.041. The capital goods category experienced a decrease in the squared sum of trade imbalances from 0.129 in 1992 to 0.060 in 2020. The highest value for this category was 0.143 in 1993, and the lowest value was 0.033 in 2008. Consumer goods saw a decrease in the squared sum of trade imbalances from 0.128 in 1992 to 0.131 in 2020. The highest value was 0.142 in 2007, and the lowest value was 0.072 in 1996. The intermediate goods category experienced a decrease in the squared sum of trade imbalances from 0.008 in 1992 to 0.007 in 2020. The highest value for this category was 0.020 in 2009. Raw materials show an overall decrease in the squared sum of trade imbalances from 0.274 in 1992 to 0.154 in 2020. The highest value for this category was 0.287 in 1993, and the lowest value was 0.080 in 2005.

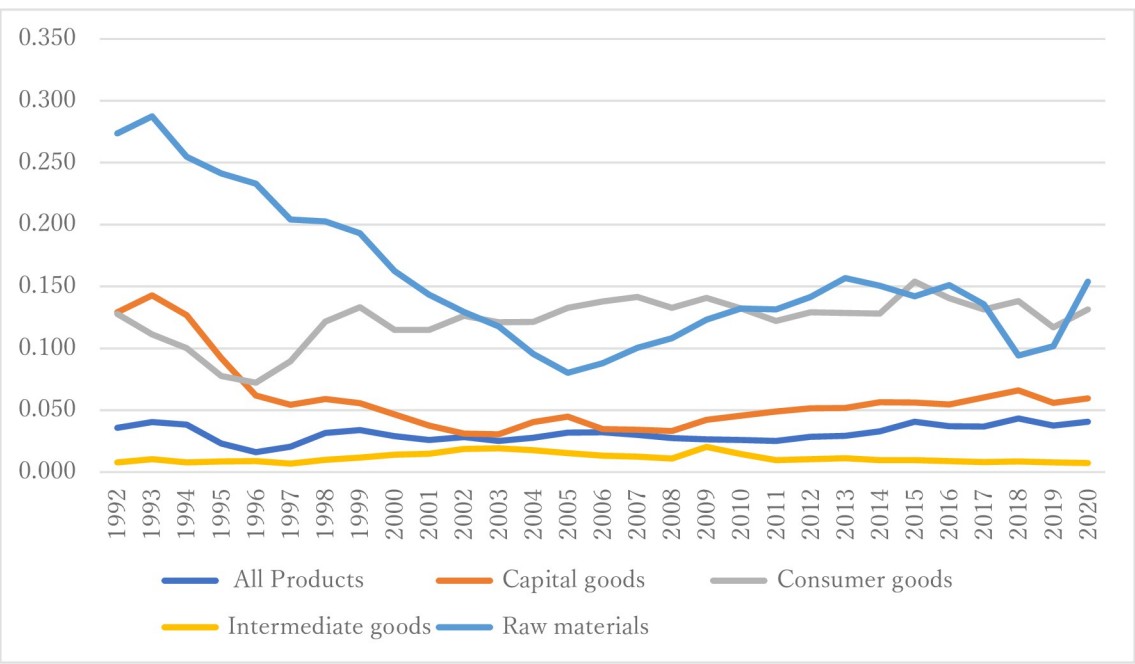

**Fig 2. Time series variation in trade between partners.**

Regarding the annual rate of change, the all products category experienced the highest positive rate of change in 2015 at 23.3%, while the highest negative rate of change occurred in 1995 at -39.8% (Table 2). The capital goods category saw the highest positive rate of change in 2004 at 32.6%, while the highest negative rate of change occurred in 1996 at -32.9%. The consumer goods category experienced the highest positive rate of change in 2015 at 20.1%, while the highest negative rate of change occurred in 1995 at -22.6%. The intermediate goods category saw the highest positive rate of change in 2009 at 87.9%, while the highest negative rate of change occurred in 1994 at -25.8%. The raw materials category experienced the highest positive rate of change in 2020 at 51.3%, while the highest negative rate of change occurred in 2004 at -18.8%.

In summary, the time-trend analysis indicates that all products have experienced an increase in squared trade imbalances, while capital goods, intermediate consumer goods, and raw materials exhibit a decreasing trend over the period from 1992 to 2020. Consumer goods, however, show a more stable trend throughout the same period. This implies that excluding all products and consumer goods, it can be said that there has been a shift towards a more reciprocal network structure in each product category. Furthermore, while reciprocity is initially high at the all-products level, it does not increase further; instead, it exhibits a slight decreasing trend.

From Table 3, the analysis of trade imbalances between 1992 and 2020 shows varying trends across different categories of products. For all products, there was an overall increase of 14.3% in squared trade imbalances, indicating a growth in trade imbalances during this period. In contrast, the capital goods category experienced a significant decrease in squared trade imbalances, with a rate of change of -53.8%, suggesting a reduction in trade imbalances for capital goods over the period. Intermediate goods also saw a decrease in squared trade imbalances, albeit at a smaller rate of -6.6%. This demonstrates a reduction in trade imbalances in the intermediate goods category during the studied period. However, the consumer goods category

**Table 2. Time series variation in trade between partners.**

| year | 1992 | 1993 | 1994 | 1995 | 1996 | 1997 | 1998 | 1999 | 2000 | 2001 | 2002 | 2003 | 2004 | 2005 | 2006 | 2007 | 2008 | 2009 | 2010 | 2011 | 2012 | 2013 | 2014 | 2015 | 2016 | 2017 | 2018 | 2019 | 2020 |
|---|---|---|---|---|---|---|---|---|---|---|---|---|---|---|---|---|---|---|---|---|---|---|---|---|---|---|---|---|---|
| All Products | 0.036 | 0.040 | 0.038 | 0.023 | 0.016 | 0.020 | 0.032 | 0.034 | 0.029 | 0.026 | 0.028 | 0.025 | 0.028 | 0.032 | 0.032 | 0.030 | 0.027 | 0.026 | 0.026 | 0.025 | 0.028 | 0.029 | 0.033 | 0.041 | 0.037 | 0.037 | 0.043 | 0.038 | 0.041 |
| Capital goods | 0.129 | 0.143 | 0.127 | 0.092 | 0.062 | 0.054 | 0.059 | 0.056 | 0.047 | 0.038 | 0.031 | 0.031 | 0.040 | 0.045 | 0.035 | 0.034 | 0.033 | 0.042 | 0.045 | 0.049 | 0.052 | 0.052 | 0.056 | 0.056 | 0.055 | 0.060 | 0.066 | 0.056 | 0.060 |
| Consumer goods | 0.128 | 0.111 | 0.100 | 0.078 | 0.072 | 0.089 | 0.122 | 0.133 | 0.115 | 0.115 | 0.126 | 0.121 | 0.121 | 0.133 | 0.138 | 0.142 | 0.133 | 0.141 | 0.132 | 0.122 | 0.129 | 0.129 | 0.128 | 0.154 | 0.140 | 0.132 | 0.138 | 0.117 | 0.131 |
| Intermediate goods | 0.008 | 0.010 | 0.008 | 0.008 | 0.009 | 0.007 | 0.010 | 0.012 | 0.014 | 0.015 | 0.019 | 0.019 | 0.018 | 0.015 | 0.013 | 0.013 | 0.011 | 0.020 | 0.014 | 0.010 | 0.010 | 0.011 | 0.010 | 0.010 | 0.009 | 0.008 | 0.009 | 0.008 | 0.007 |
| Raw materials | 0.274 | 0.287 | 0.255 | 0.241 | 0.233 | 0.204 | 0.203 | 0.193 | 0.163 | 0.143 | 0.129 | 0.118 | 0.096 | 0.080 | 0.088 | 0.100 | 0.108 | 0.123 | 0.132 | 0.131 | 0.142 | 0.157 | 0.150 | 0.142 | 0.151 | 0.135 | 0.094 | 0.102 | 0.154 |
| Annual rate of change | | | | | | | | | | | | | | | | | | | | | | | | | | | | | |
| All Products | | 12.9% | -4.7% | -39.8% | -30.4% | 26.7% | 55.2% | 7.3% | -14.4% | -10.5% | 9.0% | -11.5% | 10.5% | 15.3% | 0.9% | -6.6% | -9.2% | -3.5% | -1.4% | -2.9% | 12.5% | 3.5% | 12.2% | 23.3% | -8.9% | -0.9% | 17.9% | -13.1% | 8.3% |
| Capital goods | | 10.7% | -11.2% | -27.4% | -32.9% | -12.1% | 8.6% | -5.5% | -16.5% | -19.4% | -17.1% | -2.0% | 32.6% | 10.5% | -22.4% | -1.4% | -3.0% | 27.2% | 7.7% | 7.4% | 5.6% | 0.6% | 8.7% | -0.4% | -2.6% | 10.3% | 9.4% | -15.2% | 6.5% |
| Consumer goods | | -13.0% | -9.9% | -22.6% | -7.0% | 24.0% | 35.8% | 9.7% | -13.8% | -0.1% | 9.9% | -4.0% | 0.1% | 9.3% | 4.0% | 2.6% | -6.2% | 5.9% | -6.0% | -7.8% | 5.7% | -0.3% | -0.4% | 20.1% | -8.8% | -6.3% | 5.1% | -15.5% | 12.4% |
| Intermediate goods | | 33.2% | -25.8% | 9.0% | 3.9% | -22.3% | 45.1% | 17.2% | 20.5% | 5.2% | 26.9% | 2.8% | -8.6% | -13.0% | -13.0% | -6.3% | -13.4% | 87.9% | -29.1% | -32.4% | 7.3% | 8.0% | -15.6% | 1.5% | -9.8% | -7.9% | 6.9% | -8.2% | -6.8% |
| Raw materials | | 5.0% | -11.4% | -5.2% | -3.5% | -12.4% | -0.8% | -4.7% | -15.8% | -11.9% | -9.7% | -9.0% | -18.8% | -16.1% | 9.6% | 14.1% | 7.7% | 14.1% | 7.3% | -0.7% | 7.8% | 10.8% | -4.0% | -5.6% | 6.3% | -10.3% | -30.4% | 8.0% | 51.3% |

displayed a positive rate of change, with an 2.8% increase in squared trade imbalances, indicating a slight growth in trade imbalances for consumer goods over the period. Lastly, the raw materials category followed a trend similar to capital goods, with a significant decrease in squared trade imbalances and a rate of change of -43.7%. This suggests that trade imbalances in raw materials were notably reduced between 1992 and 2020. In summary, the analysis shows that all products and consumer goods experienced an increase in squared trade imbalances during the period, while capital goods, intermediate goods, and raw materials all exhibited a decrease in trade imbalances from 1992 to 2020. The largest decline concomitant with 9/11 in 2001 was observed in capital goods, followed by raw materials (Fig 2). All products also significantly decreased, ranking third. Consumer goods showed little decrease in linkages compared to the previous year in a wide range of links. However, intermediate goods slightly increased in 2001 (S1–S5 Tables).

Following the analysis of the time series variation in trade balance between partners, the study observes an overall increase in squared trade imbalances for all products and consumer goods, but a decrease in the squared trade imbalances for capital goods, intermediate goods, and raw materials. This could potentially be due to economic factors such as changes in production capacities and consumption habits of different regions, or shifts in international policies and agreements affecting trade. It also indicates a shift towards a more reciprocal network structure in each product category excluding all products and consumer goods, which may suggest an increasingly interdependent global economy. However, it is important to note that this trend does not necessarily equate to a fairer or more balanced trade system, as the imbalance is initially low at the all-products level, but does not decrease further; instead, it exhibits a slight increasing trend (Table 3).

## 4.2. Visualization of the time series progression of network structure

**4.2.1. All products.** From Figs 3 and 4, as for all products, the international trade system initially involved only J and U in a closed relationship. C entered the scene by exporting to U, eventually expanding its exports to J as well. Subsequently, J began exporting to C. Over time, the relationship between J and U lost its bidirectional nature, turning into an export-only relationship from J. K later joined the network through exports to C. Judging from Fig 3, the center shifted from U to C in 2007.

For detailed information on the transition of the central position in the network structure, please refer to the S1–S5 Tables.

From 1992 to 2000, J consistently dominated the outflow of products, as well as U. In the inflow of products, both J and U played dominant roles during this period. Between 2001 and 2004, C emerged as the dominant region for both outflows and inflows of products, with J and U remaining significant contributors. K became moderately dominant in outflows from 2007, but not in inflows. In 2014, J's dominance in outflows declined, but it remained consistent in inflows. U' dominance decreased in both outflows and inflows. From 2016 to 2020, J regained its presence in outflows. In inflows, J maintained its dominance, while U exhibited its presence

**Table 3. Rate of change 2020/1992.**

| | |
|---|---|
| All products | 14.3% |
| Capital goods | -53.8% |
| Consumer goods | 2.8% |
| Intermediate goods | -6.6% |
| Raw materials | -43.7% |

| | 1992 | 1994 | 1996 | 1998 | 2000 | 2001 | | 2004 | 2006 | 2008 | | 2010 | 2012 | 2014 | 2016 | 2018 | 2020 |
|---|---|---|---|---|---|---|---|---|---|---|---|---|---|---|---|---|---|
| all products | | UJ | | | U | UJ | J | | JC | | | | | C | | | |
| capital goods | | | UJ | | | | UJC | | | | C | | | | | | |
| consumer goods | U | | UJ | | | | | | UJC | | | | | | | | C |
| intermediate goods | | | J | | | | | JC | | | | C | | | | | |
| Raw materials | | | UJ | | | U | UC | U | C | UC | U | UC | | | U | | |

**Fig 3. Transition of hub from 1992 to 2020.**

during this period. Throughout the entire period, I and K had minimal or no impact on product flows.

In summary, the dominant regions in product flows over the period from 1992 to 2020 included J, which was consistently dominant in both outflows and inflows, with a brief decline in outflows in 2015 and 2018; U, which was consistently dominant in inflows and present in outflows from 1992 to 2003; C, which was dominant in both outflows and inflows from 2002 to 2020; K, which was moderately dominant in outflows from 2007 to 2019 but not in inflows; and I, which had minimal to no impact on product flows throughout the period. Moreover, the data indicates that there has been some increase in trade flow links in the early 2000s, particularly for regions J, U, and C. Additionally, the total trade flow links have also increased

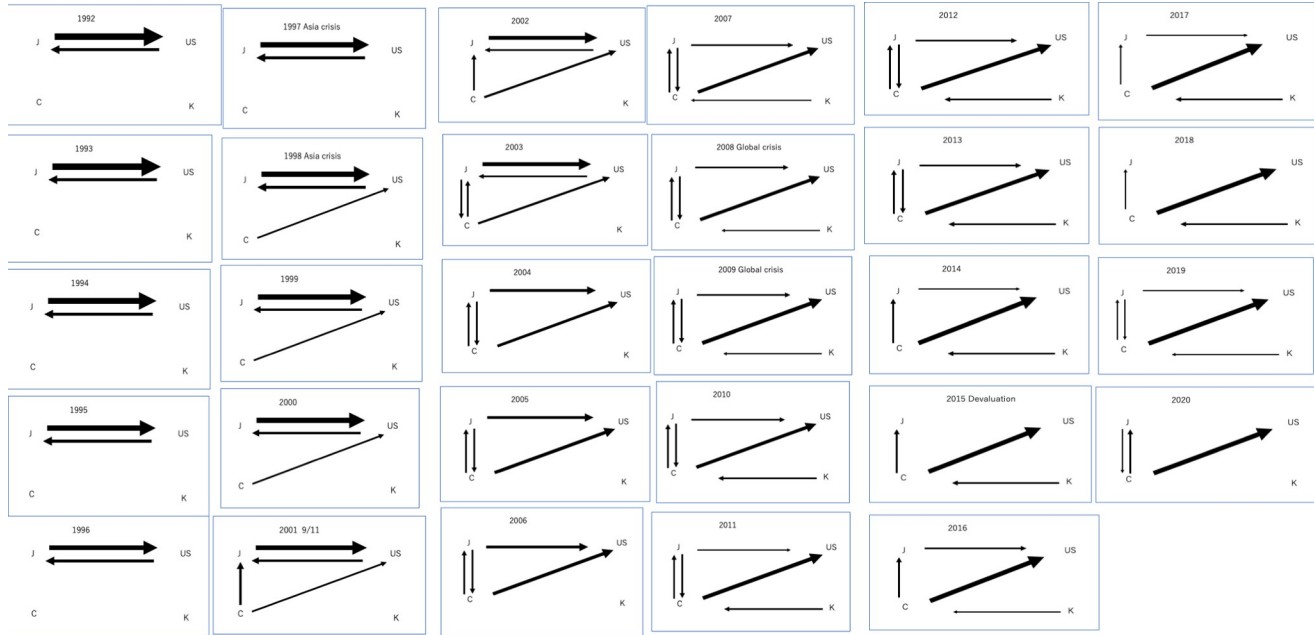

**Fig 4.** a Visualization of the time series progression of network structure for all products:1992–2001. b Visualization of the time series progression of network structure for all products:2002–2011. c Visualization of the time series progression of network structure for all products:2012–2020.

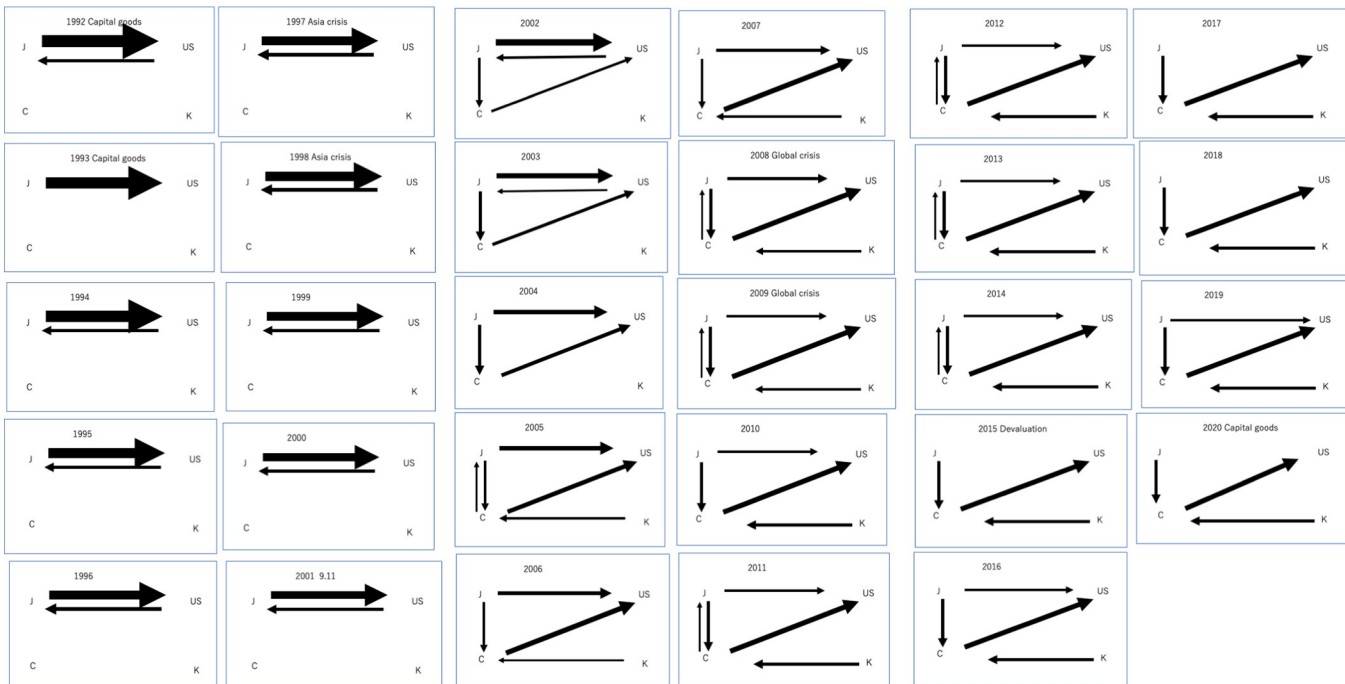

**Fig 5.** a Visualization of the time series progression of network structure for capital goods:1992–2001. b Visualization of the time series progression of network structure for capital goods:2002–2011. c Visualization of the time series progression of network structure for capital goods:2012–2020.

over time for these regions. In contrast, the K and I regions experienced less change, with the K region showing a slight increase in outgoing trade flow links from the mid-to-late 2000s, and the I region remaining stable throughout the entire period.

**4.2.2. Capital goods.** From Fig 5, in the global capital goods flow, the initial system involved only J and U, forming a closed network. Around 2002, C entered the network, supplying U and becoming an export destination for J. However, in 2001, the system temporarily reverted to a closed state involving only J and U before returning to its previous state the following year. From 2003, U transitioned into a pure importer, and K emerged as an exporter to C in 2005. During the 2015 devaluation, the system was reduced to three main flows, consisting of C exporting to U and J and K exporting to C. J's exports to U resumed sporadically every few years. Judging from Fig 3, the center shifted from U to C in 2005.

From 1992 to 2001, U and J consistently dominated the outflow of capital goods as well as the inflow. Between 2002 and 2005, C emerged as the dominant region for both outflows and inflows of capital goods. K became moderately dominant in outflows from 2015 but not in inflows. U maintained a strong dominance in inflows during this period.

In summary, the dominant economies in capital goods flows are U and J, mostly dominant in both outflows and inflows from 1992 to 2020, with a brief slight decline in inflows from 2002; C, dominant in both outflows and inflows from 2002; K, moderately dominant in outflows from 2005 but not in inflows; and I, with minimal to no impact on capital goods flows throughout the period.

**4.2.3. Consumer goods.** From Fig 6, in the global consumer goods flow, J dominated the outflow of consumer goods from 1992 to 1997, while U and C maintained a moderate presence. During this period, J and U were dominant in the inflow of consumer goods. However, from 1998 to 2020, C emerged as the dominant nation for outflows of consumer goods, with J remaining a significant contributor. U experienced a decline in dominance for outflows but

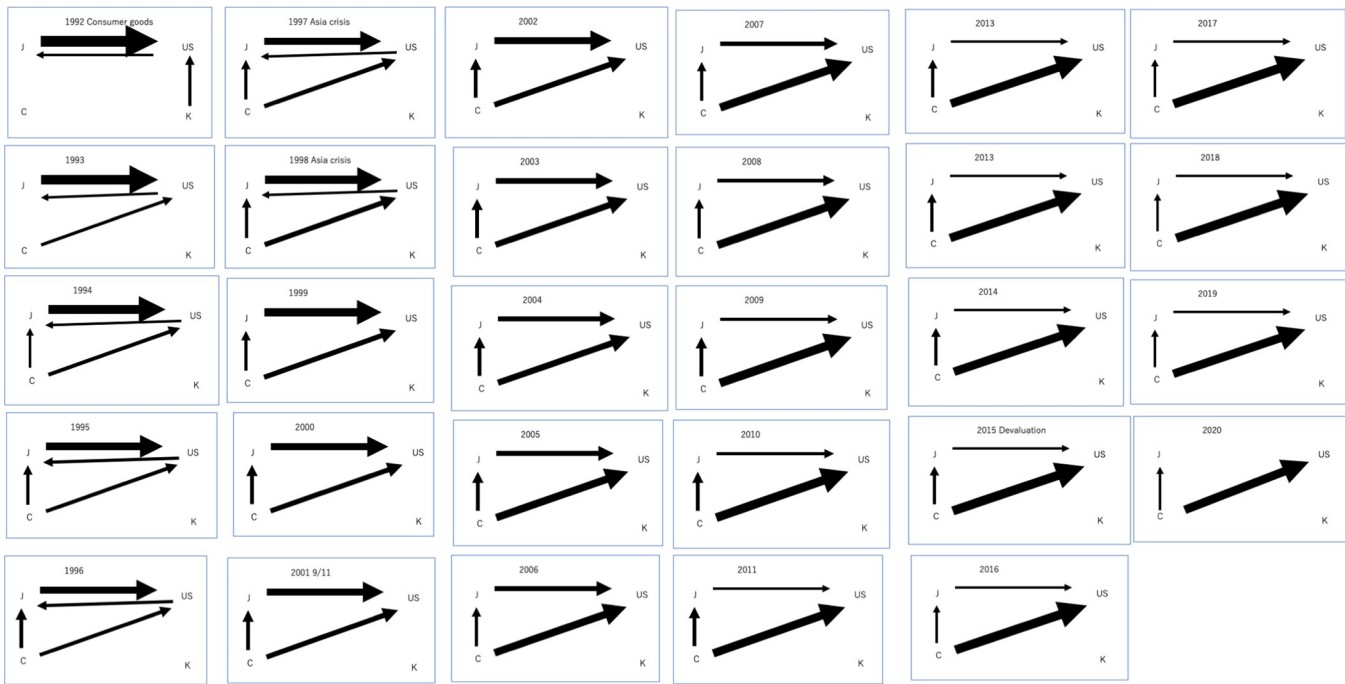

**Fig 6.** a Visualization of the time series progression of network structure for consumer goods:1992–2001. b Visualization of the time series progression of network structure for consumer goods:2002–2011. c Visualization of the time series progression of network structure for consumer goods:2012–2020.

continued to dominate inflows. U, C, and J played major roles from1999 to 2020, which shows multipolarization of the network.

Throughout the entire period, K had minimal impact on consumer goods flows, with some presence in outflows only in 1992, while I had no impact on consumer goods flows at all. In summary, the dominant countries in consumer goods flows include J, which maintained a consistent presence in both outflows and inflows from 1992 to 2020; U maintained a consistent dominance in inflow from 1992 to 2020 and outflow from1992 to 1998; C, dominant in outflows from 1993 to 2020 and no presence in inflows throughout the period; K, with minimal impact and some presence in outflows only in 1992; and I, with no impact on consumer goods flows throughout the period.

**4.2.4. Intermediate goods.** From Fig 7, in comparison to other product networks, the most diverse network has developed early on, and it has exhibited a tendency to become increasingly diversified over time. However, since 2015, C has increasingly become a hub, resulting in a lack of links between countries other than C. From 1992 to 2003, J dominated the outflow of intermediate goods, while Regions U and K maintained a moderate presence. During this period, U and C were consistently present in the inflow of intermediate goods, with J and K exhibiting a moderate presence. Judging from Fig 3, the center shifted from U to C in 2006.

From 2004 to 2020, C emerged as the dominant economy for outflows of intermediate goods, with J continuing to be a significant contributor. U's dominance in outflows declined from 2005, but it maintained a consistent presence in inflows. K sustained a consistent presence in both outflows and inflows from 1995. Throughout the entire period, I had no impact on the outflow of intermediate goods, while it gained some presence in inflows starting from 2015.

In conclusion, the data shows that the trade flow links for intermediate goods across all regions are relatively stable, with some fluctuations over time. J experienced a decrease in

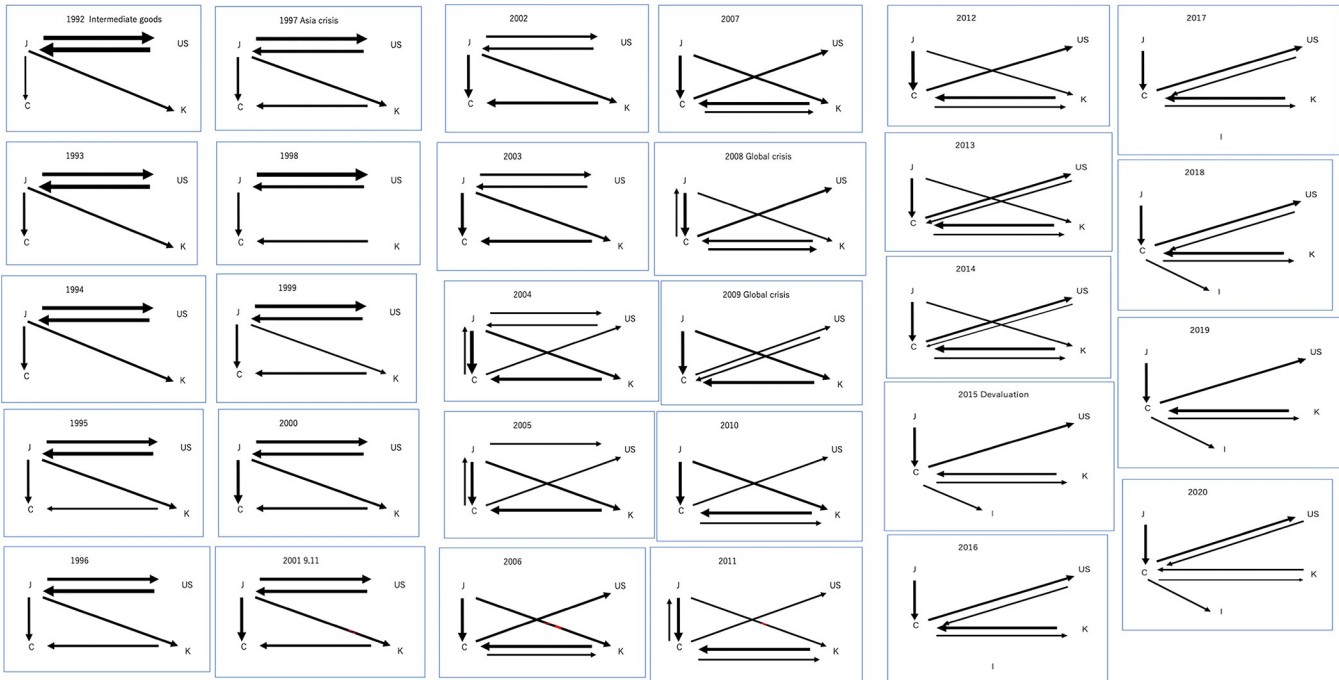

**Fig 7.** a Visualization of the time series progression of network structure for intermediate goods:1992–2001. b Visualization of the time series progression of network structure for intermediate goods:2002–2011. c Visualization of the time series progression of network structure for intermediate goods:2012–2020.

outgoing trade flow links in 2015. U experienced a decrease in outgoing trade flow links in 2005. C and K experienced increases in both incoming and outgoing trade flow links over time. I experienced an increase in incoming trade flow links in 2015.

**4.2.5. Raw materials.** From Fig 8, when examining the network from the supply side, U is overwhelmingly at the center, with its supply destinations becoming increasingly diversified over time. From 1992 to 2020, U consistently dominates the outflow of raw materials, with a slight increase in outflows starting from 2001. C has a moderate presence in raw materials outflows from 1992 to 2007, but experiences a significant decline after that. I gains some presence in the outflow of raw materials starting from 2005 but remains relatively small compared to other countries. J and K have no impact on the outflow of raw materials throughout the period.

In contrast, J consistently dominates the inflow of raw materials throughout the entire period. C and K also dominate the inflows starting from 2001 and 1992, respectively. U has no inflows of raw materials, while I has a very limited presence, with inflows only in 2018 and 2019. In summary, J and U are the dominant regions in raw materials flows, with J focused on inflows and U focused on outflows. C, K, and I play varying roles in the flows, with C and K having a moderate presence in inflows and outflows and I maintaining a limited presence in both. In summary, the inflows from U has been gradually diversified.

**4.2.6. Total number of links.** From Table 4, in each product category, it can be said that the nation with the highest total value plays a central role. From 1992 to 2020, excluding intermediate goods, the country with the highest total value in all product categories is U, indicating that it plays a central role. However, from 2008 to 2020, excluding consumer goods and raw materials, the category with the highest total value in all product categories is C, signifying that it has assumed a central role (Table 5). In the realm of consumer goods, the number of arrows for U, C, and J are roughly equal, suggesting a state of tri-polar equilibrium.

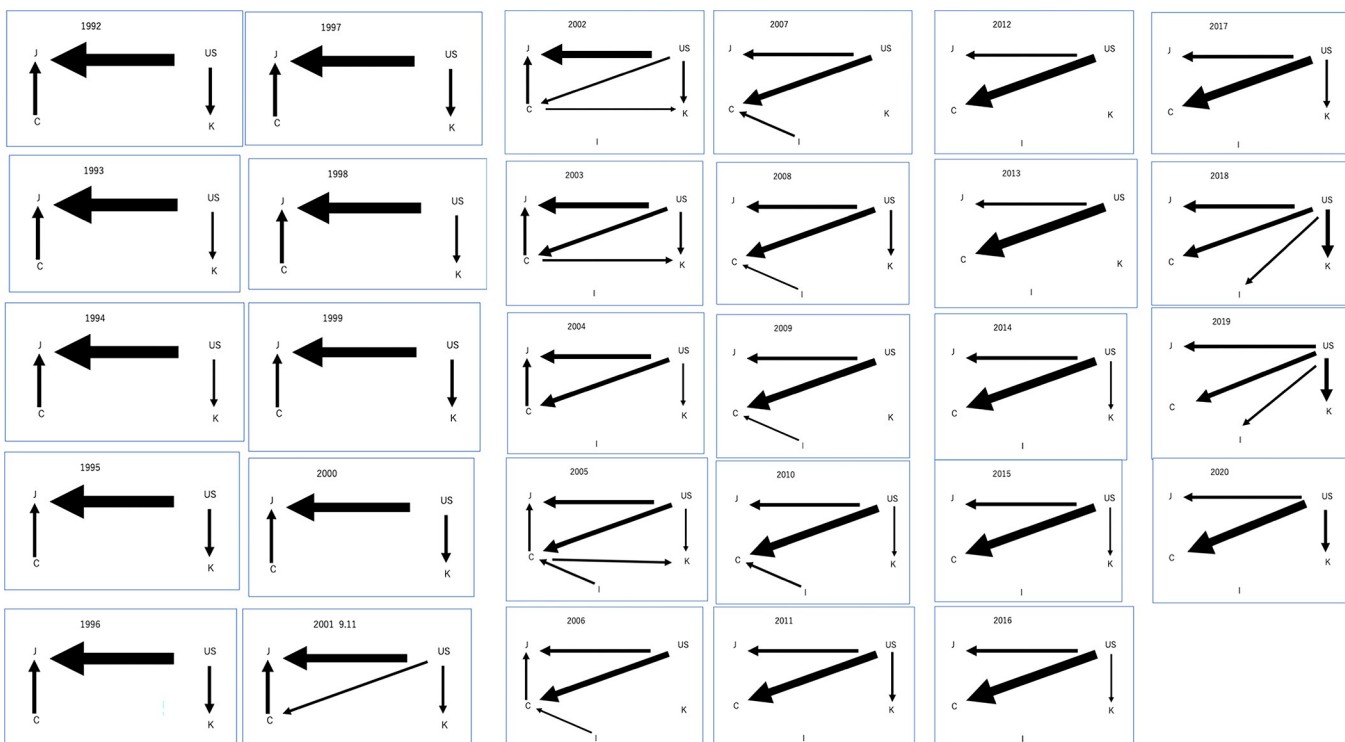

**Fig 8.** a Visualization of the time series progression of network structure for raw materials:1992–2001. b Visualization of the time series progression of network structure for raw materials:2002–2011. c Visualization of the time series progression of network structure for raw materials:2012–2020.

Consequently, the center of the network has shifted from U to C in all product categories, except for consumer goods and raw materials. In 2001, concomitant with 9/11, raw materials experienced a substantial increase in the number of major arrows, rising from 6 to 8. This shift can be construed as a reactive measure to prevent unipolar concentration, culminating in a distributed flow across the entire network. Conversely, no changes were observed in the count of major arrows for other product categories.

From Fig 3, the center of the network has shifted from U to C in 2007 for all products, 2005 for capital goods, 2006 for intermediate goods. There was no shift for consumer goods and raw materials. When we delve into the product category analysis, the study finds differing dominance patterns across product categories and countries over time. For instance, while J was consistently significant in outflows and inflows for all products, U and C fluctuated in dominance over time. This could be due to changing economic landscapes and industrial capacities in these economies. Similarly, for capital goods, we observe the shifting dominance of regions

**Table 4. Count of arrows with a standard deviation of 55 or greater, 1992–2020.**

|  | All products | Capital goods | Consumer goods | Intermediate goods | Raw materials |
|---|---|---|---|---|---|
| U | 220 | 57 | 79 | 76 | 65 |
| C | 195 | 40 | 58 | 109 | 28 |
| K | 102 | 37 | 19 | 58 | 25 |
| J | 199 | 14 | 84 | 76 | 39 |
| I | 10 | 0 | 0 | 8 | 2 |
| TOTAL | 726 | 148 | 240 | 327 | 159 |

**Table 5. Count of arrows with a standard deviation of 55 or greater, 2008–2020.**

|  | All products | Capital goods | Consumer goods | Intermediate goods | Raw materials |
|---|---|---|---|---|---|
| U | 23 | 22 | 25 | 20 | 38 |
| C | 46 | 45 | 26 | 64 | 16 |
| K | 12 | 13 | 0 | 32 | 10 |
| J | 31 | 28 | 25 | 22 | 13 |
| I | 0 | 0 | 0 | 4 | 5 |
| TOTAL | 112 | 108 | 76 | 142 | 82 |

such as U and J to C over the course of the study period. This could potentially reflect industrialization and development trends, particularly in emerging economies such as C. It would be interesting to further investigate how these changes align with socioeconomic transformations in these nations.

On the other hand, the consumer goods flow displays a multi-polar network with U, C, and J playing major roles. This could potentially be influenced by factors such as consumer demand, industrial capacity, and trade policies in these regions. The stable nature of this network could suggest an established and resilient trade system. The intermediate goods category exhibits a diverse and increasingly diversified network, possibly reflecting the interconnected nature of global supply chains. This is especially pertinent given the centrality of intermediate goods in the manufacturing process. However, the study also observes a consolidation of hub status in C since 2015, which could indicate a potential vulnerability in the network and warrants further investigation. For raw materials, the study observes U's dominance in outflows and J's dominance in inflows, which could possibly be tied to their respective natural resource endowments, production capacities, and consumption needs. This highlights the critical role of resource allocation in global trade.

Overall, it was observed that the network center shifted from U to C in 2007 for all products, in 2005 for capital goods, in 2006 for intermediate goods. The study's finding that the shift happened in 2007 for all products is in agreement with the previous research [12]. Moreover, our study found that the point at which shifts occurred varies by product category (Fig 3). These changes may have broader implications. For example, the shift from U to C as the central node in all product categories (except for consumer goods and raw materials) could reflect broader geopolitical shifts and has potential implications for global economic stability and development. Further, the increase in the number of major arrows for raw materials in 2001, concomitant with 9/11, could suggest an impact of global events on trade patterns. A deeper understanding of these trends and their drivers is crucial to the formation of trade policies.

## 4.3. Hierarchical clustering

As indicated in Fig 9, the event of 9/11 is marked as a pivotal point, segregating the entire dataset into two distinct subsets 1992–2000 and 2001–2020. The subset encompassing the years 1992–2000 further divides into two clusters, 1992 and 1993–2000. Then 1993–2000 divides into 1993–1994 and 1995–2000. Moreover 1995–2000 divides into 1995–1997 and 1998–2000, in tandem with the Asian financial crisis. Conversely, the 2001–2020 subset segregates into 2001–2009 and 2010–2020. 2001–2009 bifurcates into 2001–2003 and 2004–2009. 2004–2009 subdivides into 2004–2007 and 2008–2009 clusters, corresponding with the global financial crisis. The 2010–2020 subset bifurcates into two clusters, 2010–2014 and 2015–2020, during the period of China's currency devaluation. In conclusion, the 2015–2020 subset subdivides

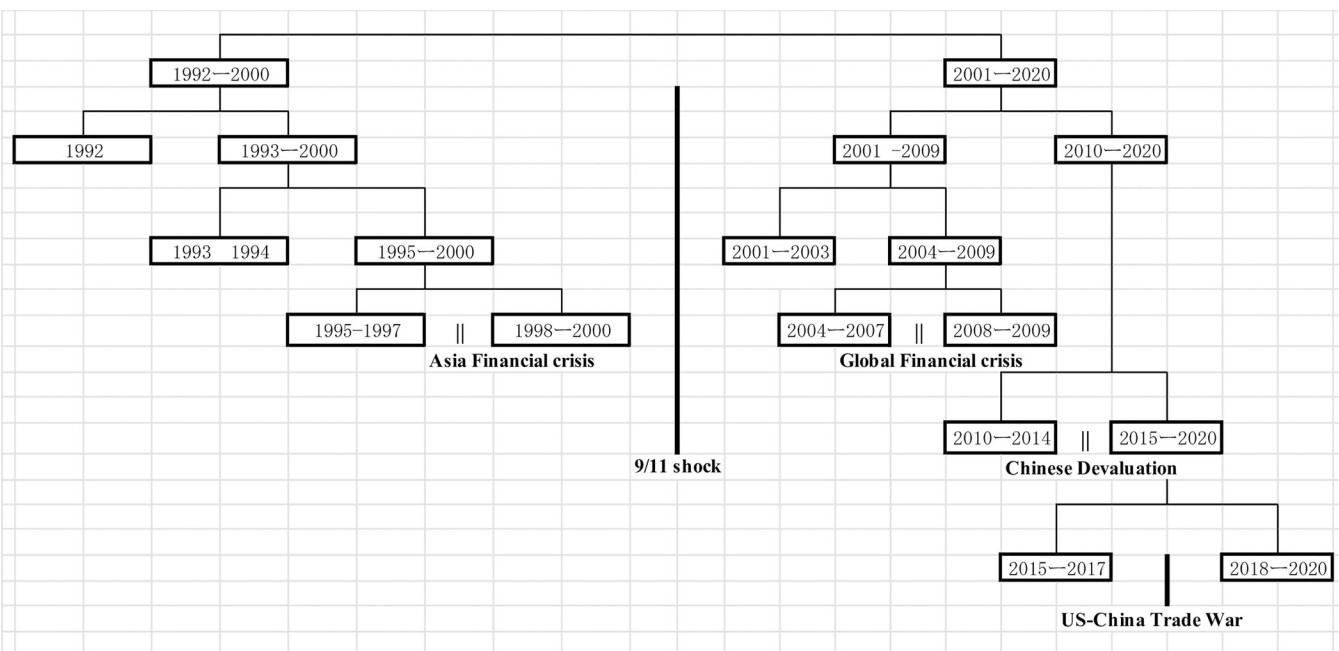

**Fig 9. Dendrogam of hierarchical clustering analysis.**

into the 2015–2017 cluster and the 2018–2020 cluster concurrently with the instigation of the US-China trade war, provoked by Trump's inauguration. The cluster analysis illuminates that the 9/11 attacks had the most substantial influence on the network structure, succeeded by the global financial crisis. The third considerable impact came from the Asian financial crisis and China's currency devaluation, while the US-China trade war during the Trump administration had the least significant effect on the structure thus far. Of the nine divisions observed in the diagram, five align with crises, suggesting that around half of the significant modifications in network structure occur during incidents like 9/11 and economic crises.

Fig 9 demonstrates the alterations in the network structure post various crises, providing crucial insights into the dynamics and resilience of global trade. The occurrence of considerable changes in the network structure during such incidents indicates a substantial vulnerability of global trade to these crises. However, the impact of these crises on the network structure varies.

The 9/11 event had the most significant effect, suggesting that global geopolitical events could disrupt trade relationships and reshape trade flows potentially due to political realignments, shifts in regional stability, or interruptions to global supply chains. Additionally, the disruptions post 9/11 were unparalleled, possibly prompting nations to diversify trade links, leading to substantial alterations in the network structure. However, while assessing the 9/11 aftermath, it is crucial to consider that China formally joined the World Trade Organization (WTO) that same year. Disentangling the long-term impacts of 9/11 from those originating from China's WTO accession can be a complex process.

In contrast, the US-China trade war under Trump's administration had the least influence on the network structure, indicating that trade disputes, although disruptive, might not significantly impact the global trade network as a whole. This could also suggest adaptability in the network, with other nations compensating for such disputes and thereby preserving the overall structure. The Asian financial crisis, the global financial crisis, and China's currency devaluation corresponded with considerable network structure alterations. These economic crises

could induce shifts in regional economic landscapes, affecting trade flows. For instance, a financial crisis might weaken a country's economic stance, curbing its import-export capabilities, and leading to the renegotiation of trade deals.

It is essential, however, to note that the correlations identified in this study do not necessarily imply causation, warranting further research to conclusively determine these crises' impact on global trade. In summary, this analysis underscores the dynamic nature of global trade, significantly influenced by major geopolitical and economic crises. Recognizing these dynamics and vulnerabilities is imperative for policymakers and businesses, enabling them to anticipate changes in global trade flows and devise strategies to mitigate potential risks.

## 5. Conclusions and discussions

### 5.1. Conclusions

This study elucidates the dynamic landscape of international trade networks by scrutinizing the evolution of trade relationships between major exporting economies—U, C, I, J, and K—from 1992 to 2020. The key conclusions are threefold:

First, the network structure has progressively become more balanced for most product categories, as evidenced by diminishing squared trade imbalances for capital goods, intermediate goods, and raw materials, while such imbalances have expanded for all products and consumer goods. Second, there has been a shift in the network's nucleus from U to C for all product categories excluding consumer goods and raw materials, underscoring C's burgeoning global influence in tandem with its economic growth and industrial expansion. Third, global events and crises, including the 9/11 attacks and the global financial crisis, wield a substantial influence on the international trade network structure. This emphasizes the necessity of factoring in geopolitical events and economic shocks in the analysis of trade dynamics. These findings carry significant implications for international trade policy and economic development. Policymakers need to recognize the evolving nature of trade networks and the influence of global events in molding trade relationships when crafting strategies to stimulate growth and bolster resilience. Businesses should prepare for potential shifts in trade dynamics and adjust their strategies accordingly.

Distinct from the prevailing research, this study introduces a unique methodological approach and offers novel insights. Unlike Garlaschelli and Loffredo's [10] focus on link reciprocity and the 2015 study's [11] application of multi-regional input-output data, our approach uses the sum of squared trade imbalances to assess network reciprocity and implements clustering analysis to probe trade pattern changes. Unlike the 2022 paper [12] that employed an eigenvector centrality measure, we leverage a time series visualization to explore network structure dynamics. This innovative perspective reveals a divergence in reciprocity trends across product categories, offers insights into shifts in the network center occurring at different times for each product category, and provides evidence of trade network susceptibility to significant crises. Essentially, our work enriches the understanding of global trade networks, offering fresh methodologies and insights into their structure and dynamics.

### 5.2. Discussions

The results of this longitudinal analysis of major exporting economies from 1992 to 2020 provide essential insights into the shifting dynamics of international trade networks. This study focused on the US, C, I, J, and K, revealing how their trading relationships have evolved over time and the impact of global events and economic crises on these networks.

**5.2.1. Shifting trade imbalances.** Our analysis of time series variation in trade between partners revealed that the squared trade imbalances for all products decreased during the

period under study, while all products exhibited an increase in squared trade imbalances. Consumer goods showed a more stable trend throughout the same period. This suggests that the network structure has become more reciprocal for most product categories, as Garlaschelli et al. [10] argued, except for all products and consumer goods. Moreover, the largest decrease in trade imbalance concomitant with 9/11 was observed in capital goods, followed by raw materials with all products, ranking third. Consumer goods showing no decrease compared to the previous year in a wide range of links and intermediate goods slightly increased. These findings shed light on the issue of the impact of 9/11 on different industries and product categories, which was left unsolved by Globerman and Storer [15]. Lastly, the decrease in squared trade imbalances for capital goods, consumer goods, and raw materials categories could be attributed to various factors, such as globalization, regional integration, and the emergence of new trading partners. The growth of global value chains and the fragmentation of production processes across countries could have also contributed to the reduction in trade imbalances.

**5.2.2. Evolution of network structure.** The visualization of the time series progression of network structure revealed how the roles of the major exporting economies have evolved over time. U, C, and J have consistently dominated the trade network, while K has gained prominence in recent years. I, on the other hand, has had minimal impact on the network throughout the period. The analysis also demonstrated that the center of the network has shifted from U to C for all product categories, except for consumer goods and raw materials. This change could be attributed to C's rapid economic growth, industrial expansion, and increasing global influence. The finding that C has a moderate presence in raw materials outflows from 1992 to 2007, but experiences a significant decline after that is in accordance with the perspective depicted by Elobeid et.al [16],who argue that US-C trade tensions have led to a significant decrease in the exports of US agricultural commodities to C, which has in turn affected the global agricultural market. What is more, it was found that at 9/11, raw materials experienced a substantial increase in the number of major arrows but no changes were observed in the count of major arrows for other product categories. This transformation can be interpreted as a response designed to mitigate the risk of unipolar concentration, ultimately leading to a more evenly distributed flow throughout the entire network. This finding resolves a part of the unresolved issue in the study by Aggarwal & Wu and Globerman & Storer [15, 17] regarding how the 9/11 attacks affected various industrial sectors and goods.

**5.2.3. Impact of crises on trade networks.** The innovative approach of targeting 29 years from 1992 to 2020 for hierarchical clustering analysis and grouping years with similar characteristics has yielded highly intriguing results. They highlighted the significant impact of global events and crises on the international trade network structure. The 9/11 attacks, the Asian financial crisis, the global financial crisis, China's currency devaluation, and the US-C trade war during the Trump administration were identified as critical events shaping the network. The 9/11 attacks had the most profound impact on the network structure, followed by the global financial crisis. The result that approximately half of the substantial alterations in network structure transpire during incidents such as 9/11 and economic crises is somewhat contradictory to the argument by Makinen [18] that the impact of 9/11 on trade was relatively small and short-lived. On the contrary, it could be argued that it left a long-lasting imprint on the trade network structure. Moreover, this finding underscores the importance of considering geopolitical events and economic shocks in understanding the evolution of international trade networks, as is pointed out by Lane et al. [19]. Policymakers and businesses should be prepared for the potential consequences of such events on trade relationships and devise strategies to mitigate their impacts.

## 5.3. Policy recommendations

Based on the aforementioned analysis and findings, it seems feasible to propose the following policy recommendations.

1. Promotion of Balanced Trade: Given the trend towards more reciprocal network structures, policymakers could consider encouraging practices that nurture this development. This could involve incentives for balanced trading and pursuing trade agreements that uphold equality and mutual benefit. Such measures could promote economic stability and more equitable international relations, though their effects should be closely monitored.

2. Adjustment of Trade Focus: The shift of the trade network center from U to C suggests that modifying trade policies and strategies could be beneficial. Actions might include bolstering diplomatic ties, understanding and catering to C's market needs, and encouraging domestic industries to align with C's industrial expansion.

3. Strengthening Economic Resilience and Consideration of Global Events: Considering the influence of global events and crises on international trade network structures, policies aimed at enhancing economic resilience and considering geopolitical factors should be contemplated. This could include implementing risk assessment and mitigation measures and diversifying trade partnerships. Incorporating geopolitical risk analysis into trade policy development could help anticipate shifts in trade dynamics, enabling timely and effective responses to evolving global scenarios.

These recommendations are proposed with a view to bolstering economic stability, ensuring balanced trade relations, and enhancing resilience against potential shocks. They are based on current findings and should be regularly reassessed in light of new data or changing circumstances.

## 5.4. Limitations and future research

This study has some limitations that should be acknowledged. First, the analysis focused on only five major exporting economies, and the inclusion of other significant trading nations could have provided a more comprehensive understanding of the shifting dynamics in international trade networks. Second, the study period ended in 2020, and more recent developments, such as the ongoing COVID-19 pandemic, have not been considered. The pandemic has had a substantial impact on global trade and could have further implications for the structure of international trade networks.

Future research should consider expanding the scope of the study to include additional countries and extend the period of analysis to cover recent events, such as the COVID-19 pandemic. Additionally, researchers could explore the impact of regional trade agreements, such as the European Union, the African Continental Free Trade Area, and the Regional Comprehensive Economic Partnership, on the evolution of trade networks. Lastly, investigating the potential effects of emerging technologies, such as digitalization and automation, on international trade networks would provide valuable insights into the future of global trade.

## Supporting information

**S1 Table. Evolution of the centralities in trade network: All products.**
(DOCX)

**S2 Table. Evolution of the centralities in trade network: Capital goods.**
(DOCX)

**S3 Table. Evolution of the centralities in trade network: Consumer goods.**
(DOCX)

**S4 Table. Evolution of the centralities in trade network: Intermediate goods.**
(DOCX)

**S5 Table. Evolution of the centralities in trade network: Raw materials.**
(DOCX)

## Acknowledgments

I would like to express my heartfelt gratitude to my wife, Ayumi Yazawa, and my son, Tokio Yazawa, for their continuous support of my research activities over a long period of time. Additionally, I am deeply grateful to Dr. Hee-Hyun Nam, Dr. Kenichi Kawai, who serves as the head of the department, and Dr. Masashi Takaki, the dean of the faculty. I also wish to thank my other colleagues in the Faculty of International Business Management at Beppu University.

## Author Contributions

**Conceptualization:** Nobuo Yazawa.

**Data curation:** Nobuo Yazawa.

**Formal analysis:** Nobuo Yazawa.

**Funding acquisition:** Nobuo Yazawa.

**Investigation:** Nobuo Yazawa.

**Methodology:** Nobuo Yazawa.

**Project administration:** Nobuo Yazawa.

**Resources:** Nobuo Yazawa.

**Software:** Nobuo Yazawa.

**Supervision:** Nobuo Yazawa.

**Validation:** Nobuo Yazawa.

**Visualization:** Nobuo Yazawa.

**Writing – original draft:** Nobuo Yazawa.

**Writing – review & editing:** Nobuo Yazawa.

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
