## [Decision Letter · Decision Letter 0]

2 May 2023

PONE-D-23-09683Shifting Dynamics in International Trade Networks: A Longitudinal Analysis of Major Exporting Economies (1992-2020): A case of the US, China, India, Japan and South KoreaPLOS ONE

Dear Dr. Yazawa,

Thank you for submitting your manuscript to PLOS ONE. After careful consideration, we feel that it has merit but does not fully meet PLOS ONE’s publication criteria as it currently stands. Therefore, we invite you to submit a revised version of the manuscript that addresses the points raised during the review process.

We look forward to receiving your revised manuscript.

Kind regards,

Magdalena Radulescu

Academic Editor

PLOS ONE

Journal Requirements:

Additional Editor Comments:

Dear authors,

Based on the reviewers' reports, we decided major revision for your submission. See the reports listed below. Please consider very carefull each suggestion, address all comments and elaborate a response letter for reviewers point by point.

Best regards,

Magdalena Radulescu

Associate Editor Plos One

Reviewers' comments:

Reviewer's Responses to Questions

**Comments to the Author**

1. Is the manuscript technically sound, and do the data support the conclusions?

Reviewer #1: No

Reviewer #2: Yes

2. Has the statistical analysis been performed appropriately and rigorously? 

Reviewer #1: No

Reviewer #2: Yes

3. Have the authors made all data underlying the findings in their manuscript fully available?

Reviewer #1: Yes

Reviewer #2: Yes

4. Is the manuscript presented in an intelligible fashion and written in standard English?

Reviewer #1: Yes

Reviewer #2: Yes

5. Review Comments to the Author

Reviewer #1: REFEREE'S REPORT ON

"Shifting Dynamics in International Trade Networks: A Longitudinal Analysis of Major Exporting Economies (1992-2020): A case of the US, China, India, Japan and South Korea"

Comments: The authors studied “shifting dynamics in international trade networks” in US, China, India, Japan and South Korea. This paper needs improvement as has been listed below:

1.Abstract

The policy proposal of the study's originality and importance should be written in this section.

Explaining the method and variable in one sentence would be sufficient.

The policy proposal of the study's originality and importance should be written in this section.

2.Introduction

The main motivation of the study should be explained correctly in this section.

The study's importance, purpose and theoretical framework should be discussed in detail.

Theoretical explanations regarding the main topic is insufficient.

The general plan of the study should be written at the end of the section.

3.Literature

It is very important to approach literature studies critically. However, a literature review was not conducted in the study. I suggest creating a literature table.

The difference of the study from the literature and its contribution to the literature should be explained under this title

4.Data and Methods

Analyzes and the findings have not been adequately discussed.

You must explain why you chose US, China, India, Japan and South Korea and the dates 1992-2020.

You need to write the basic theory of analysis.

5.Conclusion

Policy recommendations are incomplete and inadequate.

The comparison of the study with the literature and its original contribution is not given.

I consider it appropriate to MAJOR REVISION the study for the above-mentioned reasons.

Reviewer #2: Title: Shifting Dynamics in International Trade Networks: A Longitudinal Analysis of Major Exporting Economies (1992-2020): A case of the US, China, India, Japan and South Korea

Suggestion:

The objective of this manuscript is to analyze the relationship among trade networks of exporting economies in USA, China, Japan and Korea. Data and methods are appropriate. Study has interesting findings. however, I have some concerns about this paper:

1. The author(s) should clearly present the contribution of this paper to the literature. It should be elaborated on what makes this topic an interesting research area, explaining the novelty of this research output on the subject matter.

2. The author(s) should clearly explain the empirical approaches implemented in their analysis. The author(s) provide the results of various tests without including a clear explanation of their use and why they were chosen.

3. Some arguments should be further analysed to be better supported and the ideas behind them should be developed to help the reader’s understanding (e.g. arguments included a section on the methodology and data used).

4. Some tables included and the related analysis should be revised as they are not clear for the reader to have a better understanding on the outcome of this empirical analysis.

5. Some conclusions reported cannot be fully supported by the statistical information provided. A careful revision on this matter is needed.

6. The quality of English in the paper needs to be improved.

7. Please change manuscript title, it seems rough and long.

8. Abstract need complete re-writing, please add study problem, method and results.

6. PLOS authors have the option to publish the peer review history of their article (what does this mean?). If published, this will include your full peer review and any attached files.

Reviewer #1: **Yes: **Ibrahim Cutcu

Reviewer #2: **Yes: **Umer Shahzad

---

## [Author Response · Author response to Decision Letter 0]

10 Jun 2023

In accordance with the advice from reviewer #2, we have changed the title of the paper from "Shifting dynamics in international trade networks: a longitudinal analysis of major exporting economies (1992-2020): a case of the US, China, India, Japan and South Korea" to "Dynamics of International Trade: A 30-Year Analysis of Key Exporting Nations".

---

## [Decision Letter · Decision Letter 1]

10 Jul 2023

Dynamics of International Trade: A 30-Year Analysis of Key Exporting Nations

PONE-D-23-09683R1

Dear Authors,

We’re pleased to inform you that your manuscript has been judged scientifically suitable for publication and will be formally accepted for publication once it meets all outstanding technical requirements.

Kind regards,

Magdalena Radulescu

Academic Editor

PLOS ONE

Reviewers' comments:

Reviewer's Responses to Questions

**Comments to the Author**

1. If the authors have adequately addressed your comments raised in a previous round of review and you feel that this manuscript is now acceptable for publication, you may indicate that here to bypass the “Comments to the Author” section, enter your conflict of interest statement in the “Confidential to Editor” section, and submit your "Accept" recommendation.

Reviewer #1: All comments have been addressed

Reviewer #2: All comments have been addressed

2. Is the manuscript technically sound, and do the data support the conclusions?

Reviewer #1: Yes

Reviewer #2: Yes

3. Has the statistical analysis been performed appropriately and rigorously? 

Reviewer #1: Yes

Reviewer #2: Yes

4. Have the authors made all data underlying the findings in their manuscript fully available?

Reviewer #1: Yes

Reviewer #2: Yes

5. Is the manuscript presented in an intelligible fashion and written in standard English?

Reviewer #1: Yes

Reviewer #2: Yes

6. Review Comments to the Author

Reviewer #1: REFEREE'S REPORT ON

"Shifting Dynamics in International Trade Networks: A Longitudinal Analysis of Major Exporting Economies (1992-2020): A case of the US, China, India, Japan and South Korea"

Comments: The authors studied “shifting dynamics in international trade networks” in US, China, India, Japan and South Korea. This paper needs improvement as has been listed below:

1. Abstract - All revisions done successfully.

The policy proposal of the study's originality and importance should be written in this section.

Explaining the method and variable in one sentence would be sufficient.

The policy proposal of the study's originality and importance should be written in this section.

2. Introduction - All revisions done successfully.

The main motivation of the study should be explained correctly in this section.

The study's importance, purpose and theoretical framework should be discussed in detail.

Theoretical explanations regarding the main topic is insufficient.

The general plan of the study should be written at the end of the section.

3. Literature - All revisions done successfully. Please read and check these studies. Also, please add some current studies to the literature tables.

https://doi.org/10.3390/math10152553

https://doi.org/10.1016/j.jclepro.2023.137517

It is very important to approach literature studies critically. However, a literature review was not conducted in the study. I suggest creating a literature table.

The difference of the study from the literature and its contribution to the literature should be explained under this title

4. Data and Methods - All revisions done successfully

Analyzes and the findings have not been adequately discussed.

You must explain why you chose US, China, India, Japan and South Korea and the dates 1992-2020.

You need to write the basic theory of analysis.

5. Conclusion - All revisions done successfully

Policy recommendations are incomplete and inadequate.

The comparison of the study with the literature and its original contribution is not given.

I consider it appropriate to ACCEPT the study for the above-mentioned reasons.

Reviewer #2: no further comments. All comments are addressed properly. Authors have revised the draft appropriately. I suggest to accept current draft.

7. PLOS authors have the option to publish the peer review history of their article (what does this mean?). If published, this will include your full peer review and any attached files.

Reviewer #1: **Yes: **Ibrahim Cutcu

Reviewer #2: **Yes: **Umer Shahzad

---

## [Editor Report · Acceptance letter]

19 Jul 2023

PONE-D-23-09683R1 

Dynamics of International Trade: A 30-Year Analysis of Key Exporting Nations 

Dear Dr. Yazawa:

I'm pleased to inform you that your manuscript has been deemed suitable for publication in PLOS ONE. Congratulations! Your manuscript is now with our production department. 

Kind regards, 

on behalf of

Dr. Magdalena Radulescu 

Academic Editor

PLOS ONE